# Diagnosis of Inflammatory Bowel Disease and Colorectal Cancer through Multi-View Stacked Generalization Applied on Gut Microbiome Data

**DOI:** 10.3390/diagnostics12102514

**Published:** 2022-10-17

**Authors:** Sultan Imangaliyev, Jörg Schlötterer, Folker Meyer, Christin Seifert

**Affiliations:** 1Institute for Artificial Intelligence in Medicine, University of Duisburg-Essen, 45131 Essen, Germany; 2Cancer Research Center Cologne Essen (CCCE), 45147 Essen, Germany

**Keywords:** gut microbiome, machine learning, classification, inflammatory bowel disease, colorectal cancer, stacked generalization, ensemble learning

## Abstract

Most of the microbiome studies suggest that using ensemble models such as Random Forest results in best predictive power. In this study, we empirically evaluate a more powerful ensemble learning algorithm, multi-view stacked generalization, on pediatric inflammatory bowel disease and adult colorectal cancer patients’ cohorts. We aim to check whether stacking would lead to better results compared to using a single best machine learning algorithm. Stacking achieves the best test set Average Precision (AP) on inflammatory bowel disease dataset reaching AP = 0.69, outperforming both the best base classifier (AP = 0.61) and the baseline meta learner built on top of base classifiers (AP = 0.63). On colorectal cancer dataset, the stacked classifier also outperforms (AP = 0.81) both the best base classifier (AP = 0.79) and the baseline meta learner (AP = 0.75). Stacking achieves best predictive performance on test set outperforming the best classifiers on both patient cohorts. Application of the stacking solves the issue of choosing the most appropriate machine learning algorithm by automating the model selection procedure. Clinical application of such a model is not limited to diagnosis task only, but it also can be extended to biomarker selection thanks to feature selection procedure.

## 1. Introduction

The human microbiota are connected to health and disease in many clinical applications [1,2], e.g., cancer [3] or immune-mediated inflammatory diseases [4] including e.g., inflammatory bowel disease [5]. Traditionally, the medical microbiology community used univariate statistical hypothesis tests to identify significantly different species across patient cohorts [6]. Such methods have their limitations since they neither include complex non-linear interactions among species, nor do they provide a prognostic value for a new unseen dataset [7]. Hence, most of the modern human microbiome studies rely on machine learning models to identify biomarkers of health and disease [7,8]. Machine learning also provides tools to integrate microbial data with other -omics datasets and to identify the most important species involved in health and disease [9,10]. For example, inflammatory bowel diseases such as Crohn’s disease or ulcerative colitis have similar symptoms, but their etiology and hence the treatment regimens are different [6]. Cancer treatment is another example where finding a prognostic microbial marker can be important. For instance, due to the tumor growth, the microbiota composition changes and one can detect such changes early enough to diagnose and hence intervene when treatment has the best effect on patients [11].

The choice of the machine learning algorithm for microbial and clinical data-driven problems is methodologically and computationally challenging. Since both microbial and clinical features are represented as tabular data, specialized neural network models e.g., Convolutional Neural Networks [12,13,14] are not directly suitable on such datasets. In terms of predictive performance, due to the so-called “no free lunch” theorem there is no universally best algorithm which would perform equally well across all datasets [15]. Therefore, most of the microbiome studies related to looking for solution to this challenge are empirical and often apply a large suite of models on microbial data one-by-one with the aim of finding a model which performs the best [16,17,18,19]. Such studies often suggest that using ensemble models such as Random Forest or Gradient Boosted Decision Trees results in best predictive power. However, this is still a heuristic choice and even application of such complex models individually might still lead to suboptimal results.

Moreover, the application of machine learning on practical microbiology problems must be done with great care, because the consequences of wrong decisions based on model predictions can harm patients during treatment. For example, it has been shown that black-box models like Neural Networks can make accurate predictions even when they exploit clinically undesired data patterns [20,21]. Therefore, model interpretability becomes an important issue next to the model accuracy. It is known that there is a trade-off between model interpretability and model accuracy [7]. For example, complex black-box models such as Extreme Gradient Boosting [22] or Random Forest [23] are accurate but not interpretable, whilst simple models like Logistic Regression are interpretable, but often slightly less accurate [17]. To understand how the model makes its predictions and consequently increase trust in the model, *Explainable* (also sometimes interchangeably called *Intelligible* or *Interpretable*) machine learning models [24] should be used in biomedical applications involving microbiome data.

In summary, a clinically applicable machine learning diagnostic model must be (a) accurate, (b) explainable, (c) capable of integrating multiple data sources and d) reasonably efficient in terms of computation time. Generalized stacking [25] (often called stacking for the sake of simplicity) satisfies these requirements and solves the problem of selecting the most suitable machine learning model. Like Random Forest, stacking is also an ensemble learning algorithm, but instead of aggregating weak learners, i.e., decision trees, stacking combines the output of strong learners. Those strong learners make prediction errors on different subsets of samples, i.e., stacking directly benefits from a diversity of classifiers in terms of their predictions. Each of the base learners is akin a judge in a jury committee, which sees something unseen by other of his/her colleagues, but collectively using a higher level judge’s help, entire committee achieves best judgement. The output of the base learners is not combined by simple averaging, but learnt using a so-called meta learner. Meta learning has been shown to improve predictive power, because the meta learner combines strengths of each base learner and compensates their weaknesses [26,27]. Additionally, the flexible approach of stacking allows to train base learners on different subsets of features stemming from various data views, resulting in a multi-view learning setting [28]. Moreover, stacking delivers a computational trade-off between running a single powerful machine learning algorithm (hence achieving possibly suboptimal result) and running virtually all possible algorithms one-by one (hence not achievable on practice).

In this study, we evaluate stacking on two different patient cohorts, a pediatric inflammatory bowel disease cohort [29,30] and adult patients of colorectal cancer [31]. We demonstrate that stacking leads to better result (Test Average Precision = 0.69 on inflammatory bowel disease dataset and Test Average Precision = 0.81 on colorectal cancer dataset) than using a single best machine learning algorithm, despite that each dataset poses its unique challenges from both medical and machine learning aspects. On the one hand, classifying two diseases such as Crohn’s disease and ulcerative colitis is expected to be a more difficult task rather than comparing healthy patients and patients diagnosed with cancer. On the other hand, microbiota classification of newly-onset inflammatory bowel disease in treatment-naive children should be easier than classification of adult cancer patients who have undergone various treatments during their disease progression. Comparing results on both datasets, we assess reasons why stacking can be an alternative to applying a single model. We also evaluate empirically how base learners are mutually similar to each other with respect to predictions as well as how multi-view settings can potentially lead to an improved predictive performance. For model interpretability, we also provide feature importance values retrieved from both the meta learner and the entire stacking pipeline.

## 2. Materials and Methods

### 2.1. Datasets and Preprocessing

Table 1 provides an overview of basic characteristics of both datasets, including number of features included in both clinical and microbial views as well as number of unique values on genus taxonomic level in microbial view.

The 16s rRNA microbial data features and clinical patient data features for inflammatory bowel disease patient’s cohort dataset were retrieved using the *MicrobeDS* R-library [32]. Only samples from a pediatric RISK cohort were included, excluding any patients who had any prior treatment using steroids and/or antibiotics. Among this subset, only samples of patients with ulcerative colitis and Crohn’s disease were included. To prevent overoptimistic model performance and prevent training label’s leakage, we removed disease-related features which implicitly disclose patient status e.g., disease subtype, disease extent in bowel, disease duration etc. Full list of removed features is provided in Appendix Table A1. Full list of included features is provided in Appendix Table A2. With this feature choice, we prevent the model learning shortcuts, e.g., that all patients diagnosed with certain subtype of ulcerative colitis are indeed patients diagnosed with ulcerative colitis. Moreover, from a clinical viewpoint, diagnosing a patient for whom the disease subtype is known becomes a trivial task. Our final data set consists of 535 examples with 7 features for the clinical data view and 6737 features for the microbiology view. Clinical data contains 223 missing values and no data imputation was applied. 443 samples are labeled with Crohn’s disease (negative class) and 92 labeled with ulcerative colitis (positive class).

The 16s rRNA microbial data features and clinical patient data features for colorectal cancer were retrieved from the GitHub repository [33] referred to by the authors of the original paper [31]. Only samples of patients with colorectal cancer and healthy controls were included, excluding all patients with adenoma. To prevent training labels’ leakage, disease-related features e.g., history of cancer, history of polyps etc., were removed. Full list of removed features is provided in Appendix Table A3. Full list of included features is provided in Appendix Table A4. As a result, the dataset prepared for colorectal cancer classification consists of 291 examples with 9 features for the clinical data view and 5982 features for the microbiology view. Clinical data contains 5 missing values and no data imputation was applied. 172 samples are labeled as healthy controls (negative class) and 119 samples are labeled as with cancer patients (positive class).

Both datasets were randomly split into 80% training and 20% test sets. To be able to assess overfitting, we held out the test set completely during model training, including the cross-validation experiments.

### 2.2. Machine Learning Model and Training Procedure

The multi-view stacked model consists of base learners which are trained independently from each other on original features per each view of the training set and a meta learner which is trained on predictions of base learners. To reduce computational time, the microbial features were filtered during training procedure. Namely, the microbial data was prepared as input for base learners by removing features which had zero variance, i.e., features with constant values. Furthermore, data was pre-filtered using ANOVA corrected by multiple test correction procedure after being power-transformed to achieve normal distribution [34]. To get a better error estimation, the training set was split into *k* folds and in *k* sequential iterations k−1 folds were used for training. During each round of stratified cross-validation, the remaining fold was used as validation set and only those validation set predictions were combined, subsequently stacked and provided as a training data to the meta learner. This procedure reduces overfitting, since meta learner is trained on predictions made on validation set, not on training set. As training features for the meta learner, the predicted probabilities of the base learners were used instead of predicted classification labels. The illustrative example of a multi-view stacked generalization framework is depicted in Figure 1.

We selected a diverse set of classifiers from different model classes as candidates for the base learners. The linear classifiers are Stochastic Gradient Descent classifier with Logistic Loss (SGD_LL), Stochastic Gradient Descent classifier with modified Huber Loss (SGD_HL) and the K-nearest Neighbors Classifier (KNN). The non-linear classifiers are Multi-layer Perceptron (MLP), Quadratic Discriminant Analysis (QDA), Random Forest (RF) and Histogram-based Gradient Boosting Classification (HGBC). Since HGBC is the only classifier in this suite which can directly be applied on data with missing values and categorical features, HGBC was applied on both microbial data (HGBC_otu) and clinical data (HGBC_clin). All other models were applied to microbial data only, because their applications to clinical data would require imputing missing values with a risk of introducing mistakes and one-hot encoding of categorical features, which would hugely increase the feature space and hence it might increase possible overfitting. Stacking provides a simple, yet powerful pipeline with reasonable development effort by combining and learning outputs of suboptimal base learners, hence we did not perform hyperparameter tuning of base learners.

As a meta learner, we chose Logistic Regression with Elastic Net regularization with exhaustive grid search hyperparameter optimization and *k*-fold cross-validation. Logistic Regression is advantageous to black-box models, because it is more interpretable and empirically it results in comparable performance to black-box models [28]. We additionally applied a soft voting procedure (SoftVote) on output of base classifiers, which returns the class label as argmax of the sum of predicted probabilities for comparison. All models were trained using Python v3.7 programming language, scikit-learn v1.0.2 package [35].

### 2.3. Performance Metrics and Handling Class Imbalance

Both of the datasets exhibit class imbalance, because the number of negative classes is larger than number of positive ones. The dataset for inflammatory bowel disease has a high class imbalance with only 17% examples labeled with a positive class, while for the dataset of colorectal cancer this number is 41%. Such imbalanced data requires specialized performance metrics, because using metrics like Area Under Receiver Operating Characteristic (AUROC) curve may lead to misleading results with over-optimistic performance evaluation [36]. To obtain a more relevant performance estimate, we used Precision-Recall (PR) curves, because they should be preferred to the Receiver Operating Characteristic (ROC) curves in classification of rare diseases [37] and their summarization as Average Precision (AP) values [38]. We report values on training, validation and test sets. For training and validation, we report the median values over the cross-validation folds. Test set values are calculated on the hold-out test set. The training set values are calculated on the entire training set using hyperparameters found during cross-validation. To estimate diversity and prediction errors of classifiers, Matthews’s Correlation Coefficient (MCC) was calculated, which was recommended as a more informative score in evaluating imbalanced binary classifications than accuracy and F1 score [39]. Unlike AP, MCC values are bound by [−1;+1], where +1 represents a perfect prediction, 0 an average random prediction and −1 an inverse prediction.

In addition to choosing performance metric appropriate to imbalanced class data, we also applied data space weighting approach, which allows obtaining a modified distribution biased towards the costly rare classes during training base learners [40]. We did not apply any under-sampling methods, because this may eliminate useful examples which might contain rare microbial species as potential disease biomarkers. Neither we applied any over-sampling methods, because they may increase the likelihood of overfitting and increase computational effort [36].

### 2.4. Model Interpretability

Conceptually, we are interested in model interpretation on two levels, i.e., on the level of meta learner features and on the level of original features. The former shows which of the base learners contributed to which extend to the final prediction. The latter shows how the model including the base learners uses the original input features for the prediction.

Firstly, the regression weights of the meta learner were retrieved and normalized by dividing by the absolute maximum value, so that the top feature importance absolute value would be equal to one, which simplifies comparison between values. These normalized values were then ranked in decreasing order and presented as a model explanation for meta learner, providing insight on relative contribution of each base classifier on final prediction. Secondly, to evaluate feature importance of the entire model, permutation feature importance values were calculated. To calculate a permutation feature importance a baseline metric is evaluated on a training dataset, i.e., features of both clinical and microbiology views. Then, a feature column is permuted and the metric is evaluated again without retraining the full model. The permutation importance is defined as the difference between the baseline metric and metric from permuting the feature column [23]. For a feature that does not contribute to the model’s decision, the model should be robust to changes in its value. For a highly important feature, we expect—on average—large changes in prediction for a data point if this feature values are changed.

## 3. Results

### 3.1. Classification of Gut Microbiota from Inflammatory Bowel Disease Patients

Figure 2 provides an overview of the performance of the stacked classifier and its independent base models. Figure 2a,b depict PR curves and the corresponding AP values of base classifiers, the SoftVote aggregation and the stacked classifier. On test set, the stacked classifier demonstrates the best performance (AP = 0.69), outperforming the best base classifier (MLP; AP = 0.61) and the simple ensemble meta learner (SoftVote; AP = 0.63). All models show signs of overfitting, their performance is worse on test set than on the training set.

Figure 2c,d depict the MCC values as heatmap. On the training set, the highest agreement among base classifiers is observed between RF and HGBC_otu (MCC = 0.75), while the lowest agreement is observed between QDA and HGBC_clin (MCC = 0.06). We observe a general low agreement between base classifiers, indicating that they learn different patterns which can be exploited by a meta learner. On test set, we observe the highest agreement between SGD_LL and SGD_HL (MCC = 0.62), and the lowest between SGD_LL and HGBC_clin (MCC = −0.08). The stacked classifier does not demonstrate the highest correlation with the training set labels reaching MCC = 0.72, but the highest correlation of its predictions with the test set labels (MCC = 0.57) outperforming best base classifier (MLP; MCC = 0.44). This indicates that the stacked classifier learns patterns that are better generalizable to unseen test data.

Figure 2e depicts the AP values of cross-validation on training and validation sets. The stacked classifier demonstrates the highest median AP on validation sets (≈0.49), outperforming all base models with QDA being the closest to random guessing (AP of 0.17). None of the models is free from overfitting with HGBC_clin having the least difference between median AP on training and validation sets. To estimate overfitting, we calculated AP difference between median AP on training and validation sets for both stacked model and all base learners split by the data view. Those median values are depicted on Appendix Figure A1.

Figure 3 shows most important features of the predictive model, with Figure 3a showing the sorted normalized regression weights of the meta learner. HGBC_clin demonstrates the highest importance value followed by SGD_HL and KNN which have roughly equal weight values of around 0.4 each. QDA classifier’s importance value though being non-zero is negligible, indicating that meta learner ignored predictions made by QDA base classifier. Figure 3b depicts the permutation feature importance values. The top 3 highest scores were assigned to the clinical features such as inflammation status, age and race with median importance values of about 0.10, 0.05 and 0.02 respectively. All other features in the top 10 list have lower feature importance scores (less than 0.02) and most of them are microbial features. Among them are microorganisms which are identified belonging to the taxonomic families *Ruminococcaceae*, *Comamonadaceae* and *Alcaligenaceae*, while two features are identified belonging to taxonomic order of *Clostridiales*.

### 3.2. Classification of Gut Microbiota from Colorectal Cancer Patients

Figure 4 shows an overview of the performance of the stacked classifier and its base models. Figure 4a,b depict PR curves and corresponding AP values of base classifiers, SoftVote aggregation and the stacked classifier. On test set, the stacked classifier demonstrates the best performance (AP = 0.81) outperforming both the best base classifier (SGD_LL; AP = 0.79) and simple ensemble meta learner (SoftVote; AP = 0.75). The best base learner performs better than a simple ensemble meta learner on test set (SoftVote; AP = 0.75 vs. SGD_LL; AP = 0.79). Overall, all models perform worse on test set compared to training set, which indicates overfitting.

Figure 4c,d depict MCC values as heatmap. The stacked classifier demonstrates the best correlation of its predictions with the test set labels reaching MCC=0.55, outperforming best base classifier (HGBC_otu; MCC = 0.43). The highest agreement can be observed between linear methods (SGD_LL vs. SGD_HL, MCC = 0.59), while KNN shows an overall low agreement with other base learners. HGBC_clin MCC values indicate lowest correlation with any other base classifier trained on microbial data often reaching negative MCC values. The mutual agreement between base classifiers is generally low, because the predictions made by base models are diverse enough to correctly predict different subsets of data.

Figure 4e depicts the AP values of cross-validation on training and validation sets. The stacked classifier demonstrates the highest median AP on validation set (≈0.83). All base models achieve lower median validation set AP, with QDA and MLP classifiers being the closest to the random AP of 0.41. None of the models is free from overfitting with HGBC_clin having the least difference between median AP on training and validation sets. To estimate overfitting, we calculated AP difference between median AP on training and validation sets for both stacked model and all base learners split by the data view. Those median values are depicted on Appendix Figure A2.

Figure 5 shows most important features of the predictive model, with Figure 5a showing sorted normalized regression weights of the meta learner. HGBC_clin demonstrates the highest importance value followed by SGD_LL and HGBC_otu which have weight values of around 0.6 and 0.5 respectively. The importance score of KNN is negative, indicating that meta learner used predictions made by KNN base classifier, but considered them as negatively correlated to the positive class probability prediction. This aligns with the finding in Figure 4c where KNN MCC values show the lowest correlation with any other base classifiers, and it indicates that the stacked classifier can learn to utilize incorrect predictions by negating their influence. Figure 5b shows the importance of input features calculated by permutation feature importance. The highest score is assigned to the patients’ age with median importance value of about 0.05. All other features in the top 10 list have lower feature importance score (less than 0.01) and most of them are microbial features. Among them are microorganisms which are identified belonging to the taxonomic families *Porphyromonadaceae*, *Bacteroidaceae*, *Prevotellaceae*, *Peptostreptococcaceae* and *Lachnospiraceae*.

## 4. Discussion

### 4.1. Stacking as a More Powerful Ensemble Method than Random Forest

The aim of this study was to investigate whether stacking would lead to better performance than a single model when applied to multi-view microbiome data. Previous similar studies [16,17,18,19] compared multiple machine learning models one-by-one and concluded that RF is often the best model in terms of predictive performance. We showed on two different real-world datasets that stacking outperforms RF in terms of generalization error, i.e., performance on unseen test data. Consistency of results among both inflammatory bowel disease and colorectal cancer datasets is particularly encouraging taking into account differences in patient cohorts and disease types. Higher performance of stacking seems logical because much like RF which combines output of multiple weak learners, stacking also combines output of multiple base learners. However, unlike RF, stacking combines the predictions of strong and diverse learners using training labels. If stacking is better than applying a single model, there must be a reason why it is not yet widely applied in microbiome community. In subsequent subsections we hypothesize reasons why stacking worked well on these two datasets and which recommendations we can conclude from our results.

### 4.2. Meta Learner’s Role in Stacking

Application of stacking on real-world datasets requires making certain choices and tricks which were referred as ’black art’ [27]. Even when stacking is applied correctly, it might still perform only as well as best base classifier [26] or with marginal improvement of about 3% compared to the best individual base classifier [41]. In our study, we achieved results higher than a single base classifier, with improvement of over 8% on test set for inflammatory bowel disease, possibly due to combining prediction probabilities rather than predicted classification labels themselves, which is in line with general recommendations [42]. Moreover, we showed that even a simple meta learner like soft voting can still result in a good performance, if base classifier probabilities are well calibrated. The study addressing issues in stacking [42] also recommends using a linear model as a meta learner, which is in line with another empirical study on microbiome [28] where Logistic Regression was recommended to aggregate predictions of base learners. For comparison, we additionally trained stacking using RF as a meta learner. PR curves for both data sets are depicted in Appendix Figure A3. RF as meta learner did not result in improvement of test AP on any of the datasets: compared to the stacked model with Logistic Regression as meta learner, test AP degraded from 0.69 to 0.56 on inflammatory bowel disease dataset (Appendix Figure A3a) and from 0.81 to 0.77 on colorectal cancer dataset (Appendix Figure A3b). Besides achieving better AP values, applying linear model like Logistic Regression is also advantageous due to better interpretability of stacking. This is helpful for understanding which base learner models contributed highest for the final model prediction. For example, we observed in Figure 3a that the meta learner ignored predictions made by QDA base classifier. Looking back at Figure 2c,d we observed that QDA had a poor predictive performance, and the agreement with other classifiers was also low. Hence, the QDA base classifier mostly made wrong predictions which the meta learner learned to ignore. This demonstrates why stacking is more preferable than average voting.

### 4.3. Diversity of Base Learner’s Role in Stacking

Another issue is the choice of base learners. Generally, it is recommended that base classifiers should be both accurate and diverse [27], though notion of diversity is rather vague and has no formal definition. In this study, we used domain knowledge by combining classifiers which are diverse based on their underlying mathematical roots, e.g., we trained linear models, ensemble models and a neural network model. The diversity of the classifiers outputs was confirmed empirically by large difference in MCC values between each model’s prediction, while their high accuracy was confirmed by similarly high MCC values w.r.t. classification labels. Based on performance of base learners, we would discourage blindly using RF as a single classifier, because it did not always result in better performance compared to other models.

The diversity of classifiers can be increased by applying base classifiers on different sets of variables or, in biomedical applications, on different -omics views. In fact, multiple microbiome studies often encourage integration of multi-omics data [8,9,10,43] in a single model. One of the previous studies [43] combined multiple -omics views using stacking, but unlike in our study, authors used exactly the same linear model across all views instead of applying different models. Both of the previous studies [30,31] from which we extracted datasets for this study, lack access to other -omics views, but they provide us access to clinical patient metadata which we successfully used as a complementary view for stacking. Previous study on Crohn’s Disease [30] did not use clinical traits data. On the contrary, the colorectal cancer study [31] used some of the clinical variables in a separately trained model. Authors of that study concluded that extra dataset can be complementary to microbial data as a source of diagnostic information. We demonstrated that stacking benefits from adding extra view for both colorectal cancer and inflammatory bowel disease datasets. MCC values, hence the mutual agreement, of base models trained on clinical data tend to be lower compared to the ones obtained from training base models on microbial data, meaning that those classifiers make errors on different subsets of samples. This improves the diversity of classifiers, hence improving overall stacking performance. The importance of adding clinical data is also indirectly confirmed by higher feature importance of the base learner trained on a clinical data. Such a diversity can potentially stem from the fact that clinical data reflects traits with long term effects on disease e.g., adaptive immune responses [6], race-related [44] or sex-related [31] differences, while microbial features reflect more dynamic traits, e.g., microbial dysbiosis [4,29] or response to treatment [11]. The performance of stacking model trained on inflammatory bowel disease can be further improved by including more clinical features. For instance, adding extra features which implicitly indicate the extent of the disease in a gut can help the model to learn differences between Crohn’s disease and ulcerative colitis. Ulcerative colitis starts at the rectum and progresses continuously through the colon and it affects only colon. Crohn’s disease is different, as it has a discontinuous pattern of spread and can affect entire gastrointestinal tract [45]. Such localization-related clinical features were explicitly excluded from the datasets in our experiments and are marked in Appendix Table A1 as ”Implicit training label leakage”. The PR curves on training and test sets of the stacking model trained on such an extended dataset are provided on Appendix Figure A6. Training AP increased to 0.93, Test AP also improved to 0.73, largely because HGBC_clin performance improved (Training AP = 0.65, Test AP = 0.48). We would generally discourage using such localization-related clinical features during training, since such model does not use complex interactions between microbial and clinical features. Diagnosis of two diseases is a challenging task and microbial traits are more interesting to discover from a research point of view [6,46]. We consider that Test AP value of 0.69 is high for such a complex classification task, because the random AP value for such an imbalanced dataset is 0.17. Hence stacking model’s performance improved more than three times compared to the one of the random predictor.

Analysis of median AP values on validation and training sets of each base learner for both datasets indicates moderate or high level of overfitting. This can also be observed while comparing PR curves between training and test sets. Higher overfitting of the base learners trained on microbial data compared to base learners trained on clinical data is expected due to higher feature numbers in microbial view. Nevertheless, on test set stacking performs the best, indicating that training meta learner on overfitted base learner predictions operates like a regularizer by assigning lower weights on models with higher generalization error, i.e., on the ones trained using microbial data. This information is usually not known beforehand and it can be discovered empirically by training models and comparing their performance values on a validation set. In a case of using a single model, data practitioner has to manually choose the best model among all base learners. However, this task is taken care of by a meta learner in stacked model. During training the stacked model, only the meta learner hyperparameters were extensively optimized, while base learners were trained using default hyperparameters. Taking into account smaller dimensionality of the dataset presented to the meta learner and simplicity of the meta-learner itself, training time of entire stacking pipeline was low and it still yielded a reasonable generalization error. Hence, stacking might be a good alternative to an extensive hyperparameter tuning of a single model when quick insights are preferable to the highest possible model performance.

Based on these conclusions, we would recommend to apply stacking as a tool to achieve increased performance by using diverse classifiers trained on diverse data views. This recommendation is empirical and we cannot guarantee that stacking is going to work better or worse on other diseases or patient cohorts, because there is no consensus on the most optimal stacking configuration [27]. However, conclusions and examples provided in this subsection suggest that stacking would perform better than a single model on complex biomedical phenomena with multiple processes involved on molecular level, e.g., aging [28], head and neck cancer [41], pregnancy [43] or protein sequence compression [47]. Such biological processes can have patterns which can be reflected on multiple levels e.g., proteomics, metabolomics, metagenomics, patient history data etc. Those patterns can be modeled using diverse set of machine learning models which are complementary to each other, i.e., they show similar performance but make different decisions for the same data points reflected by low mutual agreement scores depicted on Figure 2c,d and Figure 4c,d.

### 4.4. Overfitting Analysis

AP values depicted on Figure 2a,b and Figure 4a,b indicate moderate overfitting of the stacked model, because the Test AP values are lower than Training AP values. However, such an observation is possible only when the proper model testing is implemented, i.e., when the model is applied on unseen test set. Both of the previous studies which used those two datasets [30,31] lack any indication of using independent test sets. Inflammatory bowel disease study [30] reports average cross-validation scores on training set, while colorectal cancer study [31] reports out-of-bag scores. The latter is shown to be as an unreliable measure of the classification performance [48]. We performed extensive analysis of overfitting using cross-validation and independent test set. We found out that stacking achieves better AP on training and validation sets compared to any of the base classifiers. Main source of overfitting comes from base learners trained on microbial data rather than from ones trained on clinical data (Appendix Figure A1 and Figure A2). This might be due to the “curse of dimensionality“ because the number of features in microbial view is magnitudes higher than the number of features in clinical view (Table 1). Hence, reducing input dimensionality of the microbial view data is one of the simplest and computationally inexpensive ways to reduce overfitting. This step is essentially a feature selection method often called variable ranking or filtering [49]. We used statistical filtering, but due to large number of features a multiple test correction procedure is essential to reduce false discovery rate [50,51]. By using multiple test corrected pre-filtering procedure on microbial features, overfitting can be further decreased for both inflammatory bowel disease dataset to Training AP = 0.78; Test AP = 0.66 (Appendix Figure A4) and for colorectal cancer dataset to Training AP = 0.96; Test AP = 0.79 (Appendix Figure A5). However, such a strict univariate filtering can miss potentially important microbial features which otherwise would have been detected by multivariate base learners. Moreover, although overfitting decreased, so did the Test AP on both datasets. Therefore, such a decreased overfitting comes with the price of reduced test performance and it should be considered balancing all of the associated compromises attached to it. As mentioned in Section 4.3, the meta learner in stacking framework takes care of regularization task and it can be considered as an efficient replacement for extensive hyperparameter tuning of base learners to achieve improved performance on test set.

### 4.5. Model Interpretation and Further Examples of Stacking

As for the model interpretation, it can be done in several ways thanks to flexibility of stacking framework. We calculated feature importance values in a meta learning scale by retrieving weights of Logistic Regression used as a meta learner. This gave us insight into which of the base models were the most contributing ones. Those results confirmed “no free lunch” theorem empirically, because different base models were identified as important ones between two different datasets. We also retrieved feature importance values of the entire pipeline using permutation feature importance. Those results indicated that clinical variables are the most important ones across both datasets, despite being under-represented in terms of feature cardinality in feature space. Previous study on Crohn’s disease [30] identified a few microbial features among which microorganisms belonging to taxonomic order of *Clostridiales*. Microbes of the same order were also identified as important ones using permutation importance on stacking classifier used in our study. Direct comparison of results between previous study and our study is not possible, because the clinical research question of our study is different, i.e., comparing Crohn’s disease patients with ulcerative colitis patients instead of comparing Crohn’s disease patients with healthy patients. In previous study of colorectal cancer [31], authors found associations of microorganisms belonging to order of *Porphyromonadaceae* and *Lachnospiraceae*. In our study, we also identified a few features belonging to the same taxonomic order along with a few others such as *Bacteroidaceae*, *Prevotellaceae* and *Peptostreptococcaceae*.

In a broader sense, stacking is not a single algorithm, but rather a family of ensemble algorithms with many variants [27]. Moreover, stacking can be performed in multiple stages of modeling and it is not limited to combining predictions only. For example, stacked ensemble of learners can be used as a tool to improve feature selection [52,53], achieve better protein sequence compression [47], integrate outputs of multiple neural networks [54] or integrate input of neural network from several normalization methods [55] when applied on microbiome data.

### 4.6. Study Limitations

This study has two major limitations. First of all, it might have not achieved the best possible predictive performance on either of the datasets. Hyperparameters of base learners were not extensively optimized, because it would be computationally expensive, but most importantly because achieving highest AP on test set for each base learner was not our goal. We aimed to demonstrate that stacking performs better than application of a single machine learning algorithm and there is no need to choose one “best” model which is not possible due to “no free lunch” theorem. Instead, we showed that application of a machine learning can help to identify which other machine learning algorithms can perform best and let the practitioner focus on model interpretation. Thus, based on our findings we would recommend to apply stacking as a tool for model selection, for multi-view data integration and as an alternative to extensive hyperparameter optimization of a single model.

Another limitation is that we used only two datasets. Those datasets are very different from each other, but stacking demonstrated similar patterns on both of them. Since we aimed to get an understanding of stacking, we limited ourselves on a detailed analysis of the internal mechanics applied to two most representative datasets, providing data practitioners with tips why stacking worked well and what can we learn from it.

## 5. Conclusions

This study presents an application of the multi-view stacked machine learning model on two microbial datasets from different diseases, i.e., inflammatory bowel disease and colorectal cancer. The model combines multiple heterogeneous machine learning models and it achieves best predictive performance on the test set outperforming the best single classifier. Detailed analysis of the model provided insight on why stacking achieves such a high performance, mainly due to diversity of classifiers as well as using a meta learner which regularizes overfitted prediction labels from base learners. Thanks to the flexibility of the stacked model, combined usage of multiple views is possible and clinical data usage is encouraged as a complementary view to the microbial data.

From a practical point of view, the results of the study suggest that the stacked model partially solves the issue of choosing the most appropriate machine learning model by automating the selection using the meta learner algorithm. Results empirically confirmed the “no free lunch" theorem on two different microbial datasets with rather different patient cohorts and disease types. Clinical application is not limited to diagnosis task only, but it also can be extended to biomarker selection thanks to model interpretation using permutation feature selection procedure.

## Figures and Tables

**Figure 1 diagnostics-12-02514-f001:**
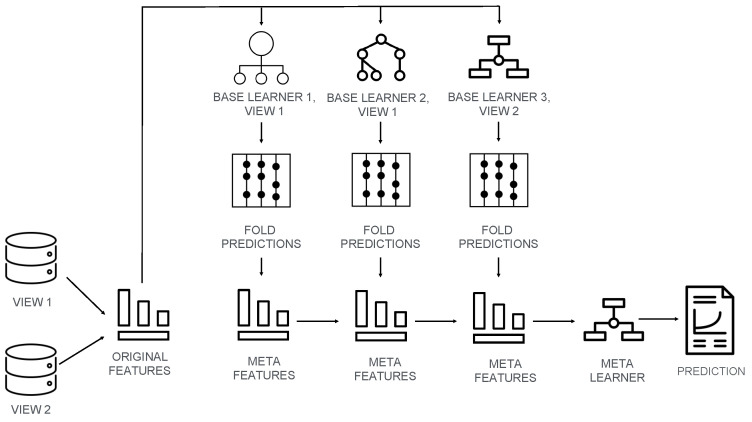
**Multi-view stacked generalization framework’s illustrative example.** Methodologically this computational framework’s example consists of three base machine learning models and a meta learner. The predictions of the base learners on validation folds are stacked together as meta features and presented for training to the meta learner model which outputs the final prediction. The illustrative example pipeline was trained on two subsets of features that allowed multi-view setting by training base learners 1 and 2 on features of view 1 and the base learner 3 on features of view 2. For the sake of simplicity, the example using only two views and three base learners is explained. In principle, number of views and base learners is not limited.

**Figure 2 diagnostics-12-02514-f002:**
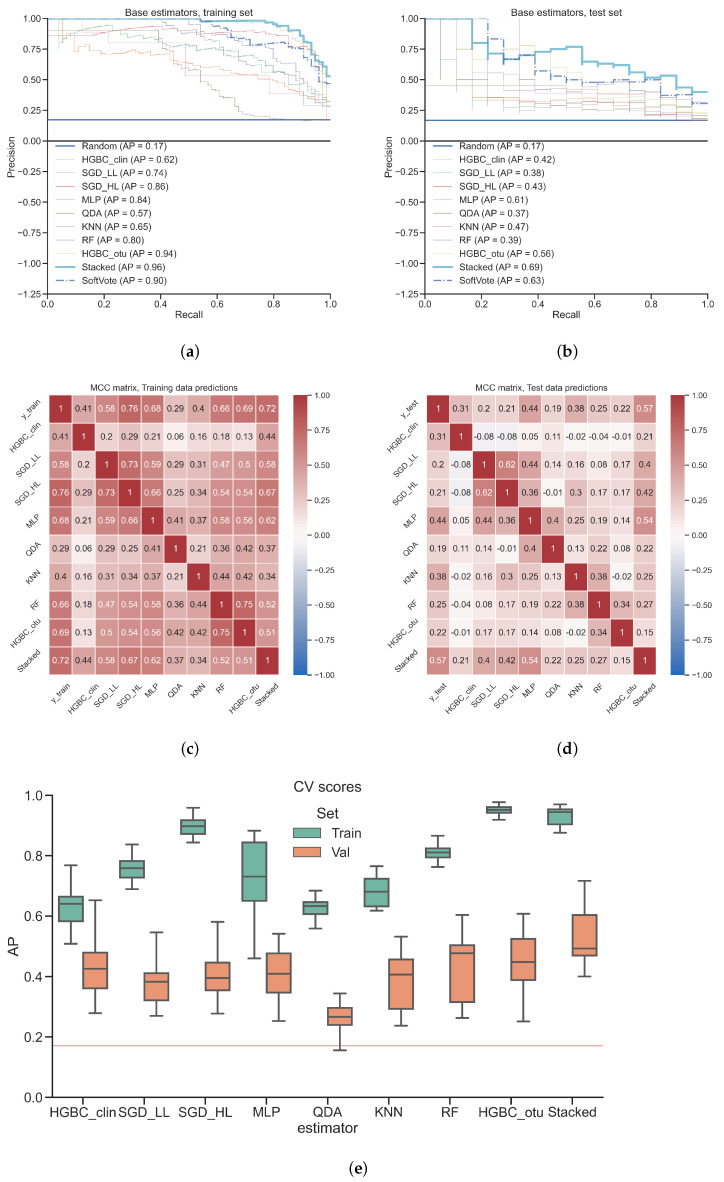
**Model performance comparison between base learners and stacked model, inflammatory bowel disease dataset.** (**a**) Precision-Recall (PR) curves and corresponding Average Precision (AP) values of base classifiers, SoftVote classifier and the stacked classifier applied on training set; (**b**) PR curves and corresponding AP values of base classifiers, SoftVote classifier and the stacked classifier applied on test set; (**c**) Matthews’s Correlation Coefficient (MCC) values matrix heatmap of base classifier predictions, classification labels and the stacked classifier predictions applied on training set; (**d**) MCC values matrix heatmap of base classifier predictions, classification labels and the stacked classifier predictions applied on test set; (**e**) AP values of each model during cross-validation on training and validation sets. Red horizontal line refers to AP value obtained from a random classifier.

**Figure 3 diagnostics-12-02514-f003:**
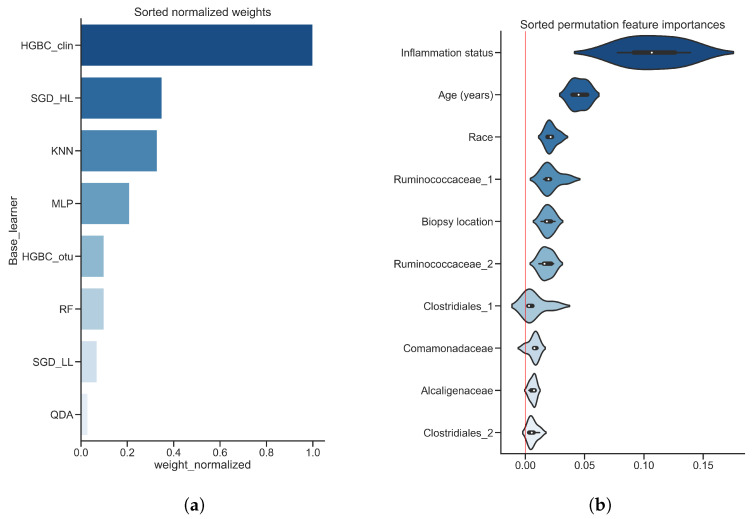
**Sorted feature importance values, inflammatory bowel disease dataset, training set.** (**a**) Normalized regression weights obtained from a meta learner used in stacked classifier; (**b**) Violin plots of permutation feature importance values obtained from a stacked classifier. Red vertical line represents zero importance value.

**Figure 4 diagnostics-12-02514-f004:**
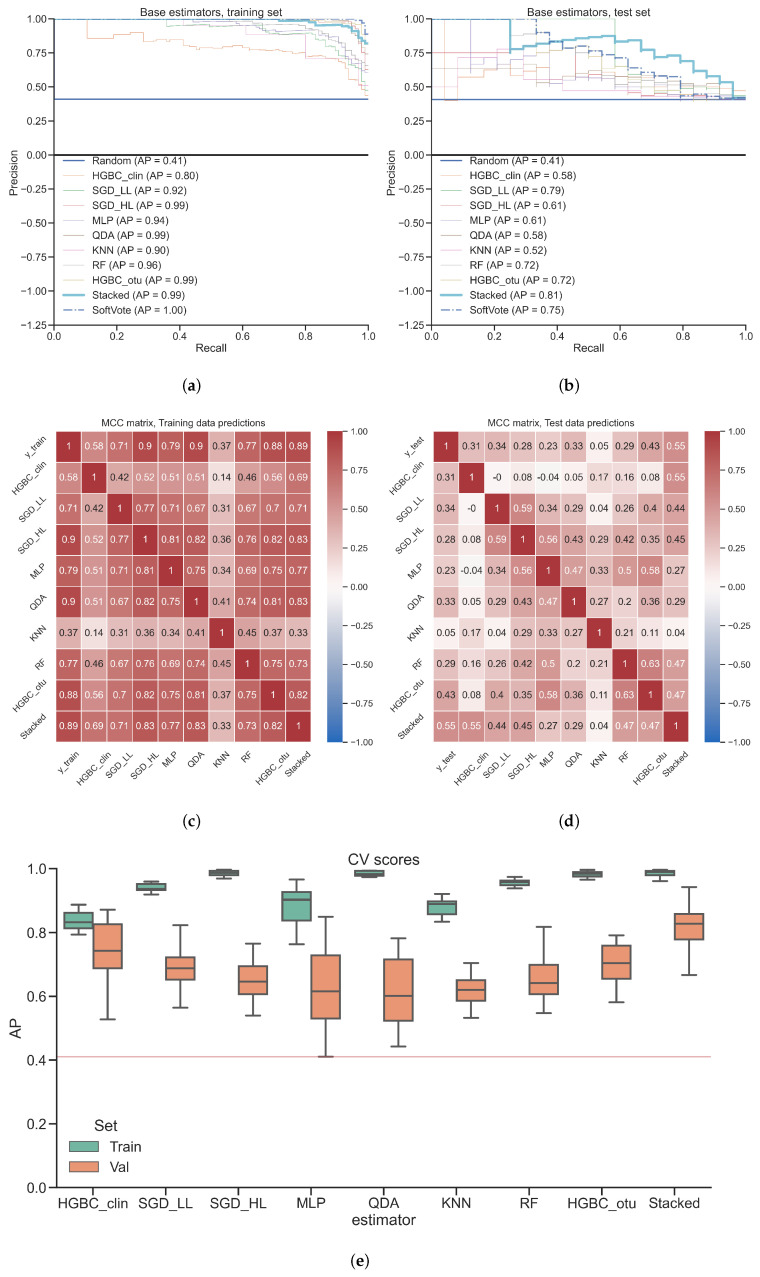
**Model performance comparison between base learners and stacked model, colorectal cancer dataset.** (**a**) PR curves and corresponding AP values of base classifiers, SoftVote classifier and the stacked classifier applied on training set; (**b**) PR curves and corresponding AP values of base classifiers, SoftVote classifier and stacked classifier applied on test set; (**c**) MCC values matrix heatmap of base classifier predictions, classification labels and the stacked classifier predictions applied on training set; (**d**) MCC values matrix heatmap of base classifier predictions, classification labels and stacked classifier predictions applied on test set; (**e**) AP values of each model during cross-validation on training and validation sets. Red horizontal line refers to AP value obtained from a random classifier.

**Figure 5 diagnostics-12-02514-f005:**
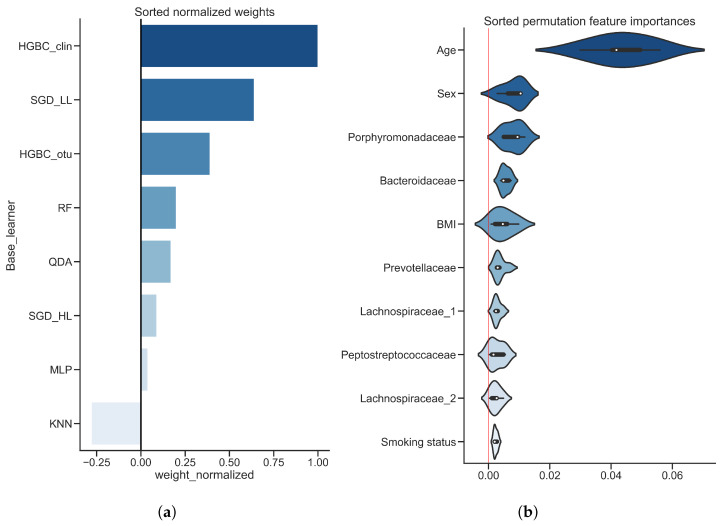
**Sorted feature importance values, colorectal cancer dataset, training set.** (**a**) Normalized regression weights obtained from a meta learner used in stacked classifier; (**b**) Violin plots of permutation feature importance values obtained from a stacked classifier. Red vertical line represents zero importance value.

**Table 1 diagnostics-12-02514-t001:** Dataset characteristics for inflammatory bowel disease and colorectal cancer.

	Clinical View Features		Microbial View Features	
	# Numerical	# Categorical	# Total	# Unique Genera
Inflammatory bowel disease	1	6	6737	533
Colorectal cancer	2	7	5982	239

## Data Availability

Data and code to reproduce results of the paper are available at https://github.com/imansultan (accessed on 13 October 2022). Publicly available datasets were analyzed in this study. These data can be found here: https://github.com/twbattaglia/MicrobeDS (accessed on 13 October 2022) and https://github.com/SchlossLab/Baxter_glne007Modeling_GenomeMed_2015 (accessed on 13 October 2022).

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
