# Peer review of "Diagnosis of Inflammatory Bowel Disease and Colorectal Cancer through Multi-View Stacked Generalization Applied on Gut Microbiome Data"

_diagnostics, 2022, doi:10.3390/diagnostics12102514_

Round 1

Reviewer 1 Report

The authors completed an interesting work and compared the competence of different classifiers in predicting inflammatory bowel disease and colorectal cancer to prove the “no free lunch” theorem. For the paper, I have several questions.

Main

1.      In figure 2 and figure 4, even the Stacking demonstrates a slightly high accuracy in the train and validation datasets, But the overfitting is much more serious, compared to the HGBC_clin. In other diseases or cohorts, can the Stacking work well? The conclusion may be over interpreted.

2.      What datasets may the stacking work better?

Minor

1.      Please explain the b_cat in table 1

2.      Please provide the full words when the abbreviation first occurs

Reviewer 2 Report

In this study, the authors evaluated a powerful ensemble learning algorithm, multi-view stacked generalization on pediatric inflammatory bowel disease and adult colorectal cancer patients’ cohorts. The authors aimed to check whether stacking would lead to better results compared to using a single best machine learning algorithm. Overall, this work is intersting. However, there are few points that are needed to concern:

1. Based on the results from Fig 1, all prediction models are found to have overfitting problem on both inflammatory bowel disease and olorectalcancer dataset datasets. Did the authors solve this issue?

2. From Comment 1, did the authors try to use under or over sampling technique?

3. Did the authors try to implement other meta-predictor, such as RF-based meta-predictor.

4. Overall, the prediction performance on the inflammatory bowel disease dataset is not quite acceptable for real application. The authors are needed to develop the effectiveness of the proposed model

5. I cannot understand the information of feature used, such as how many a  number of features for each dataset

Round 2

Reviewer 2 Report

The current version can be accepted for publication.